# Internal Quality Attributes and Sensory Characteristics of ‘Ambrosia’ Apples with Different Dry Matter Content after a Two-Week and a Ten-Week Air Storage at 1 °C

**DOI:** 10.3390/foods12010219

**Published:** 2023-01-03

**Authors:** Masoumeh Bejaei, Hao Xu

**Affiliations:** Summerland Research and Development Centre, Agriculture and Agri-Food Canada, Summerland, BC V0H 1Z0, Canada

**Keywords:** dry matter content, fruit composition, fruit quality, fruit texture, sensory evaluation, soluble solids content

## Abstract

This research was conducted to determine the compositional and textural characteristics and sensory profile of ‘Ambrosia’ apples with different dry matter content (DMC) as estimated using a Felix-750 Produce Quality Meter (Felix Instruments Inc., Camas, WA, USA). Fruits were harvested from a commercial orchard in Cawston and an experimental field in Summerland Research and Development Centre (SuRDC) in British Columbia, Canada, when the average absorbance difference index/coefficient of fruit skin *δ*Absorbance (*δ*A) dropped under 0.45 ± 0.10. DMC levels were estimated after harvest at the blush/background transition zone for fruit categorization on 300 fruits from each location. Fruits were coded with an individual number and grouped in different DMC categories. The distribution of the estimated DMC levels obtained from two locations was different. The results indicate that DMC levels were strongly and positively correlated with the soluble solids content (SSC) of the fruit (r = 0.81). Sensory evaluations also demonstrated that apples in the lowest DMC category (12.5% ± 0.5 from Cawston) were considered the least sweet apples with the least overall flavour quality by panellists compared to the apples from the other DMC categories included in the sensory evaluations from the two locations. Panellists also perceived less-than-expected “fresh apple” and “tropical” flavours but more-than-expected “no flavour” and “bland” off flavour from the lowest-DMC-category apples. The non-destructive DMC measurements show a potential to be used to sort apples for SSC, sweetness and flavour; nevertheless, they were not related to firmness or textural attributes.

## 1. Introduction

Fruit quality perceptions of consumers are influenced by both intrinsic and extrinsic characteristics of the fruit, and intrinsic attributes of the fruit such as its texture, flavour and appearance play important roles in consumers’ purchase decision-making process [1,2]. The impacts of genetic factors, agricultural practices and environmental factors on intrinsic fruit quality characteristics have been explored in the past, and multiple instrumental measurements have been developed to evaluate the fruit quality at different stages of production and storage [3]. Fruit dry matter content (DMC) has been considered as one of the fruit quality indicators [4,5,6] that is influenced by different factors such as crop load, fruit size and water and nutrient supply [6,7].

Non-destructive tools have been developed for the fruit maturity and/or quality assessments during different stages of production, harvest and storage. The measurement of the chlorophyll content of fruit skin [8,9] and the use of colorimeter and spectroscopy measurements to estimate different fruit quality attributes [10,11,12] are examples of non-destructive fruit quality assessment tools. These instruments can be very useful in evaluating the quality of products if their applications are cost-effective and if the collected data correlate well with the human perception of the fruit quality. As a result, it is important to measure the quality of the fruit using compositional/instrumental measurements and sensory evaluations at the same time to identify if the changes recorded using instruments are detectable by human senses. Then, the researchers can decide if there is a merit in the application of a specific quality measurement tool in sorting the quality of the fruit destined for commercial markets.

The non-destructive methods have been explored and proposed to measure the fruit DMC as a fruit quality indicator [12,13,14,15,16]. The DMC levels of apples have been shown to be associated with their firmness, flavour and consumer acceptance [5]. Consumer research data can demonstrate consumers’ liking or their preferences of a product, but they cannot be used to identify the sensory attributes that play a role in shaping those preferences. To the best of the authors’ knowledge, there is very limited information available on the potential influence of the DMC levels on the sensory characteristics of the fruit in the literature. As a result, the main objective of this study was to investigate the relationships between the DMC levels of ‘Ambrosia’ apples and their compositional, textural and sensory characteristics to explore the possibility of the application of non-destructive DMC estimations in sorting apples. The relationship between DMC and fruit quality indicators were explored after a short two-week and a longer ten-week air storage period. Thus, the impact of the common cold storage practices on the quality of the fruit could also be studied because the relationship between the DMC levels and quality indicators may change during the storage period, as reported in previous studies [4,5,6].

## 2. Materials and Methods

### 2.1. Fruit Characteristics and Compositional and Textural Measurements

The fruits of ‘Ambrosia’ were sampled from a commercial orchard in Cawston (49.2° N, 119.7° W; “Cawston”) and an experimental trial in Summerland Research and Development Centre (49.6° N, 119.6° W; “SuRDC”), British Columbia, in the middle of September 2021 when the average absorbance difference index/coefficient (*δ*Absorbance at 670–720 nm, *δ*A) of fruit skin dropped under 0.45 ± 0.10 (Apple DA meter, Sinteleia, Bologna, Italy) [8]. On each location, 300 marketable fruits were randomly picked from various locations in the canopy of 30 trees of ‘Ambrosia’ on Malling 9 rootstock in order to obtain a sample pool that represented a range of DMC levels.

DMC levels were estimated using a Felix-750 Produce Quality Meter (Felix Instruments Inc., Camas, WA, USA) [1,15,17], at the blush/background transition zone on the sampled fruits based on measurements conducted after harvest for fruit categorization purposes. Compared to direct DMC measurement, which involves 72 to 96 h of oven dehydration, this non-destructive method took less than a minute and kept the samples intact, which allowed the tracking of the changes in DMC and fruit mass of the same samples overtime. 

Fruits were coded with an individual number after the measurement of their DMC levels and the individual fruit mass was recorded. Then, the skin colouration data were recorded, including: (1) the distribution of red over colour estimated based on the percent area of the skin covered by red colour; (2) the intensity of red over colour estimated based on the red colour intensity scale developed for the purposes of this study (Figure 1); and (3) ground colour in three categories (green, yellow-green and yellow).

Two-hundred seventy fruits (out of three hundred fruits) from each location were selected by removing the fruits with visible defects and blemishes (e.g., sunburns, skin cuts and stem losses). Then they were kept in a room at 22.5 ± 0.5 °C and 30% RH for 1 week to ripen, and were either stored for 2 weeks or for 10 weeks at 1 ± 0.5 °C and 72 ± 2% RH prior to sample analyses. All samples were warmed up to the room temperature (at 22.5 ± 0.5 °C and 30% RH) overnight before conducting compositional, textural and sensory tests.

Compositional and textural measurements included *δ*A of fruit skin, fruit mass, flesh firmness and fruit tissue water potential of individual fruit, and soluble solids content (SSC) and titratable acidity (TA) of pooled juice samples from three apples from the same DMC category. Table 1 shows the sample size of compositional and textural measurements per category per location at each test date (October 6 for 2-week storage and December 2 for 10-week storage). The above-mentioned measurements were repeated after a ten-week air storage on three DMC categories of each location considering the availability of fruits from each location (Table 1).

The *δ*A was measured using the delta absorbance (DA) meter. With a light-blocking foam installed around the sensor of the DA meter, the sensor was firmly faced against the longitudinal blush (red)/background (yellow) transition zone on the fruit to measure *δ*A. Using the instrumental default program for apple *δ*A measurement, one reading was taken on each of the two transition zones; two readings were averaged to represent the *δ*A of the apple. The *δ*A index indicates the values of the absorption difference between 670 nm and 720 nm, which is correlated with the chlorophyll content in the peel of the apple [9].

Two measurements of firmness were taken on the two blush/background transition zones of each fruit, using a Fruit Texture Analyzer (FTA-G25, GÜSS Manufacturing Ltd., Strand, South Africa). Each fruit was then cut into six slices; two slices per fruit of three fruits from the same DMC category were combined and juiced to prepare each juice sample. SSC (°Brix values reported in %) was measured using a portable refractometer (30P, Mettler Toledo, Columbus, OH, USA). TA was measured by titrating 75 mL of diluted juice (15 mL juice in 60 mL distilled water, D = 0.2) with 0.1 M NaOH to a pH end-point of 8.1 [18], using an OrionStar T940 titrator (Thermo Scientific, Waltham, MA, USA); the results were expressed as grams of malic acid equivalents per litre of non-diluted juice (g L^−1^). The tissue water potential of the fruit hypanthium (*Ψ*_fruit_) was measured using a WP4C potentiometer (Meter Environment, Pullman, WA, USA) as explained by Xu and Ediger [16].

### 2.2. Sensory Assessments

A preliminary study was conducted at the Sensory Evaluation and Consumer Research Program at SuRDC to determine the sensory attributes and their definitions, as well as food standards for studying the sensory characteristics of apples with different DMC levels using the quantitative descriptive analysis method [19]. Then, panel members were retrained with the ‘Ambrosia’ samples from this study and the definitions of the selected attributes, measurement scales (using the 100-unit line scales [20]) and food standards as presented in Table 2. The main sensory tests were conducted at the same time that the compositional and textural measurements were conducted: after 2 weeks and 10 weeks of air storage.

For this specific study, sensory panel judges, 12 judges in total, were recruited from SuRDC staff based on their interest, availability and previous experience, and participated in an hour refresher training, two-week and ten-week evaluation sessions. The group consisted of nine females and three males between the ages of 27 and 50. The quality of the collected data was evaluated after the completion of the sensory tests, and the data obtained from one of the judges were determined to be an outlier and removed from the dataset.

Considering the distribution of DMC categories in two locations and the availability of samples, samples from two DMC categories (13.5% ± 0.5 from Cawston and 15.5% ± 0.5 from SuRDC) were used in the training sessions. Then, samples from two DMC categories from each location, 12.5% ± 0.5 and 14.5% ± 0.5 from Cawston, and 14.5% ± 0.5 and 16.5% ± 0.5 from SuRDC, were selected for the formal sensory tests.

Two apple slices (⅛ apple) were excised from each transition zone of an apple, coded with 3-digit numbers and presented in random order (Figure 2). Duplicate evaluations per selected DMC categories were conducted in each sensory session. Thus, panellists evaluated eight samples in the two-week storage session and eight samples in the ten-week storage session (i.e., two samples from each of the four DMC categories from the two locations). The samples were presented in a randomized block design to avoid panel fatigue influence. Samples were served at 22.5 ± 0.5 °C and 30% RH.

All panellists gave their informed consent for inclusion before they participated in the study. The study was conducted in accordance with the Tri-Council Policy Statement, Ethical Conduct for Research Involving Humans, and the protocol was approved by the Human Research Ethics Committee of Agriculture and Agri-Food Canada (Amendment to 2018-F-003 Bejaei).

### 2.3. Data Analysis

The DMC categories included in the study were selected based on the availability of samples in different DMC categories from two locations. Therefore, different DMC categories from two locations could be included in the tests, and the DMC categories were nested within the location factor in the experimental design of the study.

The statistical tests reported in the current study were conducted using JMP software (JMP^®^ PRO, Version 16.2.0, SAS Institute Inc., Cary, NC, USA) and OriginPro 8.0 (OriginLab, Northampton, MA, USA). The significance level of α = 0.05 was selected, and mean, medians or LSMeans were reported with their standard errors (SE) as identified.

Different statistical tests were applied on the data considering the scales of measurements and the assumptions of statistical tests as discussed in the subsections below.

#### 2.3.1. Statistical Analyses Applied on Skin Colour and Fruit Mass Data

Two-hundred seventy apples per location were included in these tests. The fruit DMC, mass and red over colour area and intensity data were recorded in interval scales as discussed before; thus, Pearson correlation coefficients were calculated to explore the relationships among the studied variables.

The ground colour was recorded as a categorical variable; therefore, the differences in the DMC (%) among its categories were explored using an ANCOVA test considering location and ground colour as the main effects, and fruit mass as a covariate to control for the potential variation resulting from the differences in the fruit mass.

#### 2.3.2. Statistical Analyses Applied on the Data Obtained from Compositional and Instrumental Measurements

A three-way double-nested ANOVA with a two-way interaction and Tukey’s HSD post hoc tests was performed with the following fixed effects: storage duration (in two levels: two-week and ten-week), location (in two levels: Cawston and SuRDC), DMC nested within location and storage duration (represented as DMC[Location × Storage duration]; at 2 weeks, 6 levels for Cawston and 5 levels for SuRDC; at 10 weeks, 3 levels for each location; detailed levels shown in Figure 3 and Figure 4), and Storage duration × Location interaction.

In the textural dataset, 1 out of 198 samples was considered an outlier because of its heavier weight compared to the rest of the samples and was removed from analyses. The compositional dataset included 48 samples.

#### 2.3.3. Statistical Analyses Applied on the Data Obtained from Sensory Evaluations

A three-way nested ANOVA with two-way interactions and Tukey’s HSD post hoc tests were utilised to analyse the sensory test results considering the following fixed and random effects: fixed effects including storage duration (in two levels: two-week and ten-week), location (in two levels: Cawston and SuRDC), DMC nested within location (represented as DMC[Location] in two levels for each location: 12.5% ± 0.5 and 14.5% ± 0.5 for Cawston and 14.5% ± 0.5 to 16.5% ± 0.5 for SuRDC), Storage duration × Location interaction, DMC × Storage duration interaction [Location]; and random effects including judge, Judge × Location interaction, Judge × DMC[Location] interaction, and residual. The sensory dataset included 176 samples.

Differences among judges are expected in sensory evaluations; however, a significant interaction effect of the judge with other factors can be concerning [21,22] which was not observed in the current study because all interaction effects of the judge effect were non-significant.

In reporting the results of the ANOVA tests, significant main effects were reported only if the two-way interaction effects were not significant. In addition, chi-square tests were utilised to explore the independence of two categorical variables (i.e., CATA questions for fruity flavours, off flavour binomial options and DMC[Location] categories). Correlation coefficients were also explored to study relationships between the variables of interest.

## 3. Results

### 3.1. DMC, Visual Colour Evaluation and Fruit Mass Data of Individual Apples

The distribution of DMCs from two locations are presented in Table 3. The majority of the fruit from the Cawston location had a DMC range from 12.5% ± 0.5 to 14.5% ± 0.5 while the majority of the fruit from SuRDC had a DMC range from 14.5% ± 0.5 to 16.5% ± 0.5. In addition, fruit mass, red over colouration (colour area and intensity) and the distribution of the skin ground colour of the fruit from three colour categories are also presented in Table 3.

#### 3.1.1. Fruit Mass

The correlation coefficient between DMC (%) and the fruit mass of individual apples was significant (r(538) = 0.16, *p* < 0.0001). Slightly higher correlation coefficients were observed between the DMC of the apples randomly selected for the instrumental tests and their mass (r(196) = 0.35, *p* < 0.0001).

#### 3.1.2. Red over Colouration

The correlation coefficients between the red over colouration area and DMC (r(539) = 0.30, *p* < 0.0001) and the intensity of the red over colour and DMC (r(539) = 0.24, *p* < 0.0001) were both significant. In addition, they were moderately correlated with each other (DMC (r(539) = 0.47, *p* < 0.0001). Furthermore, neither the red over colouration area nor intensity was correlated with the fruit mass.

#### 3.1.3. Ground Colour

The results of the ANCOVA test indicate that fruit mass was a significant covariate in the analysis, and differences in the DMC levels of the ground colour categories from two locations were significant. Apples with yellow background colour had significantly higher DMC levels (LSMeans = 14.92, SE = 0.09) compared to the ones with green (LSMeans = 14.55, SE = 0.07) or yellow-green (LSMeans = 14.49, SE = 0.05) ground colours.

#### 3.1.4. Location

Not only did SuRDC apples have greater DMC levels than those of Cawston apples (15.73 vs. 13.45; t(527.50) = 29.35, *p* < 0.0001), but they also had larger red over colouration area (70.59% vs. 59.93%; t(490.04) = 7.16, *p* < 0.0001) and more intense red over colouration (54.08% vs. 44.93%; t(476.23) = 5.23, *p* < 0.0001). However, their fruit mass was not significantly different from those of Cawston samples (M_SuRDC_ = 200.99 and M_Cawston_ = 205.57; t(523.54) = −1.42, *p* = 0.16).

Location and ground colour variables were not independent (χ^2^ (2, N = 540) = 85.62, *p* < 0.0001). Cawston samples had more-than-expected samples with either green or yellow ground colour but fewer-than-expected samples with yellow-green ground colour. The opposite was true about the SuRDC samples.

### 3.2. Compositional and Textural Data

#### 3.2.1. δAbsorbance

Significant differences were observed in the *δ*A of apples from the two locations (F(1, 180) = 13.47, *p* = 0.0003). The *δ*A *was* higher at SuRDC (0.40 *±* 0.01) than at Cawston (0.34 ± 0.01). The *δ*A dropped from 0.43 ± 0.01 at two-week storage to 0.31 ± 0.01 after ten-week storage (F(1, 180) = 41.32, *p* < 0.0001). The *δ*A differed significantly among the DMC[Location × Storage duration] categories (F(13, 180) = 4.20, *p* < 0.0001). At Cawston, the fruits in the higher DMC categories had less chlorophyll in the fruit skin, as indicated by lower *δ*A on both test dates (Figure 3A).

#### 3.2.2. Fruit Mass for Apples Included in the Compositional and Textural Analyses

The effect of storage duration on fruit mass was significant (F (1180) = 5.8, *p* = 0.02). Fruit mass was 206.58 ± 3.16 g at two-week storage, and decreased to 195.81 ± 3.17 g at ten-week storage. There was no significant difference rendered by locations or interaction. Fruits of higher DMC categories tended to have higher fruit mass; however, the difference was not significant at *p* = 0.05 (Figure 3B).

#### 3.2.3. Flesh Firmness

In the flesh firmness, significant differences were observed in the main effect of storage duration (F(1, 149) = 8.16, *p* = 0.005) and the interaction effect of Location × Storage duration (F(1, 149) = 13.37, *p* = 0.0004). A higher firmness was found in the apples from SuRDC × two-week storage (18.20 ± 0.17 lbs) compared to fruits from the other three groups of Cawston × ten-week storage (17.39 ± 0.21 lbs), Cawston × two-week (17.24 ± 0.17 lbs) and SuRDC × ten-week storage (16.95 ± 0.21 lbs). Figure 3C shows the results of DMC[Location × Storage duration] for the flesh firmness measurements.

#### 3.2.4. Soluble Solids Content

SuRDC apples had greater SSC levels (14.43% ± 0.07%) compared to the ones from Cawston (12.61% ± 0.07%) (F(1, 31) = 311.07, *p* < 0.0001). Storage duration and interaction effects were not significant. Among the DMC[Location × Storage duration] categories, SSC increased in both locations and on both test dates by increasing DMC (F(13, 31) = 13.26, *p* < 0.0001; Figure 4A). A positive linear correlation was found between SSC and DMC (r = 0.81, *p* < 0.0001; n = 47) (Figure 5A).

#### 3.2.5. Acidity

Titratable acidity was significantly higher in SuRDC apples (3.72 ± 0.05 g L^−1^ of malic acid) than in Cawston apples (3.50 ± 0.05 g L^−1^) (F(1, 31) = 10.24, *p* = 0.03), and marginally higher in two-week stored apples (3.78 ± 0.04 g L^−1^) than in ten-week stored apples (3.44 ± 0.05 g L^−1^) (F(1, 31) = 24.21, *p* < 0.0001). Significant difference was also found in the Location × Storage duration interaction, showing higher TA concentrations in two-week stored SuRDC apples (3.82 ± 0.06 g L^−1^), two-week Cawston apples (3.75 ± 0.06 g L^−1^) and ten-week SuRDC apples (3.63 ± 0.08 g L^−1^) compared to TA in ten-week Cawston apples (3.25 ± 0.08 g L^−1^) (F(1, 31) = 5.08, *p* = 0.03). Among the DMC[Location × Storage duration] categories, the top two DMC categories from SuRDC and the top DMC category from Cawston had the highest level of TA compared to the next two DMC levels from SuRDC and the second DMC category from Cawston, which had higher TA than the last two DMC categories from Cawston (F(13, 31) = 13.27, *p* < 0.0001) (Figure 4B).

The initial pH of the juice (Appendix A) was significantly lower in Cawston apples (3.93 ± 0.03) than in SuRDC apples (4.07 ± 0.04) (F(1, 47) = 9.55, *p* = 0.003), and was significantly lower in 10-week stored apples (3.88 ± 0.01) than in two-week stored apples (4.07 ± 0.03) (F(1, 47) = 25.62, *p* < 0.001). Significant difference was found in the Location × Storage duration interaction, showing higher pH in two-week stored SuRDC apples (4.19 ± 0.04), compared to two-week stored Cawston apples (3.95 ± 0.04), ten-week stored Cawston apples (3.89 ± 0.02) and ten-week stored SuRDC apples (3.88 ± 0.02) (F(1, 47) = 11.11, *p* = 0.02). Significant difference was also found between the DMC levels of ten-week stored Cawston apples (F(2, 7) = 5.15, *p* = 0.0498).

#### 3.2.6. Fruit Flesh Water Potential

Two-week stored apples had higher *Ψ*_fruit_ (−1.80 ± 0.02 MPa) than ten-week stored apples (−1.92 ± 0.03 MPa) (F(1, 32) = 9.55, *p* = 0.004). Cawston fruits had higher *Ψ*_fruit_ (−1.77 ± 0.03 MPa) than SuRDC fruits (−1.95 ± 0.03 MPa) (F(1, 32) = 22.86, *p* < 0.0001). The highest *Ψ*_fruit_ was observed in Cawston fruits with the lowest DMC levels; the lowest *Ψ*_fruit_ was in SuRDC fruits in the top DMC categories (F(12, 32) = 2.97, *p* = 0.007; Figure 4C). A negative quadratic correlation was found between *Ψ*_fruit_ and SSC (Pearson’s r = −0.67, *p* < 0.0001; n = 47) (Figure 5B).

### 3.3. Sensory Evaluation Results

#### 3.3.1. Flesh Hardness

The model explained 43.77% of the variation in the flesh hardness data. Storage duration was the main influential factor on the flesh hardness evaluations recorded by the panellists (F(1, 128) = 13.40, *p* = 0.0004), and fruits were considered softer after storage in general (M_ten-week_ = 56.83 vs. M_two-week_ = 63.66, SE = 2.71). No significant differences were observed in the flesh hardness evaluations of apples from different DMC[Location] levels (Table 4).

#### 3.3.2. Juiciness

The model explained 25.34% of the variation in the juiciness data, and no significant differences were detected in the juiciness evaluations of the fruit from different DMC[Location] categories (Table 4).

#### 3.3.3. Overall Texture Quality

The model explained 61.45% of the variation in the overall texture quality data, and no significant differences were detected in the overall texture quality evaluations of the fruit from different DMC[Location] categories (Table 4).

#### 3.3.4. Sweetness

The model explained 65.22% of the variation in the sweetness data. The evaluation of sweetness was impacted by both location and DMC[Location] factors. Apples from SuRDC (M = 58.47, SE = 3.65) were considered to be sweeter than their Cawston (M = 53.19, SE = 3.65) counterparts (F(1, 10) = 12.66, *p* = 0.005). Greater sweetness ratings were reported for the higher DMC categories (F(2, 20) = 5.18, *p* = 0.015, Table 4).

#### 3.3.5. Tartness

The model explained 73.21% of the variation in the tartness data, and no significant differences were detected in the tartness evaluations of the fruit from different DMC[Location] categories (Table 4).

#### 3.3.6. Overall Flavour Quality

The model explained 56.26% of the variation in the overall flavour quality data. The evaluations of overall flavour quality were impacted by both location and DMC[Location] factors. Apples from SuRDC (M = 66.90, SE = 3.24) were considered to be better in flavour quality than their Cawston (M = 61.39.19, SE = 3.24) counterparts (F(1, 10) = 5.30, *p* = 0.04). This can be explained by better quality evaluations reported for the higher DMC categories (F(2, 20) = 7.11, *p* = 0.005, Table 4), and the fact that apples from SuRDC were more often in the higher DMC categories.

#### 3.3.7. Fruity Flavour

Chi-square tests of independence were performed to examine the relationships between the DMC[Location] and the fruit flavour binomial CATA choices. Fewer-than-expected panellists reported “tropical” (χ2 (df = 3, sample size N = 176) = 9.042, *p* = 0.03) and ”fresh apple” (χ2 (3, N = 176) = 17.08, *p* = 0.0007) flavours for the apples from the Cawston 12.5% ± 0.5 DMC category, while more-than-expected panellists reported “tropical” flavour for SuRDC 14.5% ± 0.5 apples. In addition, more-than-expected panellists reported ”no fruity flavour” for the apples from the Cawston 12.5% ± 0.5 DMC category (χ2 (3, N = 176) = 9.59, *p* = 0.02).

#### 3.3.8. Off Flavours

Chi-square tests of independence were used to explore the relationships between the DMC[Location] and the off flavour binomial CATA choices. More than expected panellists reported “bland” flavour for the apples from the Cawston 12.5% ± 0.5 DMC category (χ2 (3, N = 176) = 30.77, *p* < 0.0001) while fewer-than-expected panellists reported ”no undesirable flavour” for the apples from those apples (χ2 (3, N = 176) = 18.49, *p* = 0.0003).

## 4. Discussion

Apple mass was positively correlated with DMC level in the current study and similar results were reported by Saei et al. [6]. Apples with greater DMC levels also demonstrated higher red over colouration area and intensity, and they had more yellow ground colour than expected. The development of skin colour in apples is affected by different environmental factors, production practices and maturity at harvest [23,24,25,26]. As a result, the changes reported in relation to the skin colouration in the current study can be an indication of more advanced maturity levels in apples with higher DMC levels, and this was demonstrated by lower *δ*A in those apples [8]. ‘Ambrosia’ is a bi-coloured apple, and for these apples the over-colour is produced by anthocyanins (phenolic compounds) and the ground colour is developed from replacing chlorophyll with yellow pigments [24,27]. Dever et al. [28] reported that apples with more red over colouration within an apple variety were either sweeter and fruitier or sweeter and higher in pH depending on the type of the variety. This is consistent with the results from this ‘Ambrosia’ apple study—the red over coloration of larger area and greater intensity found in SuRDC apples (Table 3) was associated with higher DMC, higher SSC (Figure 4A) and higher initial pH (Appendix A).

The results indicate that the textural attributes of ‘Ambrosia’ apples explored in this study were not influenced by the DMC of the fruit. This was confirmed by both instrumental and sensory data (Figure 3C, Table 4). In contrast with the results of the current study, Saei et al. [6] reported that the firmness in ‘Royal Gala’ apples was influenced by fruit DMC. Palmer et al. [29] reported differences in the DMC levels of three apple varieties, and the early-season ‘Gala’ demonstrated a lower DMC level in the harvested fruit compared to two mid-season apple varieties (i.e., ‘Braeburn’ and ‘Fuji’). ‘Ambrosia’ is a mid-season apple variety, and the observed difference between the results reported by Saei et al. [6] and the current study in regard to the influence of DMC on texture attributes can be explained by established differences in the DMC levels of different apple varieties.

The soluble solids content was lower in fruits with lower DMC levels, especially in fruits from Cawston, as they were lower in DMC compared to those from SuRDC in general. The sensory results also indicate that the sweetness perception of panellists was lower in the lowest DMC category from Cawston (i.e., 12.5% ± 0.5 DMC) compared to other three categories. McGlone et al. [4] also reported a strong positive correlation between SSC and DMC, and Aprea et al. [30] demonstrated that sweetness was correlated positively with SSC.

The tartness perception of panellists was not significantly influenced by the DMC levels of the fruit. Nevertheless, TA compositional measurements demonstrated increases in the gram malic acid required per 100 L of the apple juice in fruits with higher DMC levels. Hampson et al. [31] also reported differences in the analytical TA measurements compared to perceive sensory sourness in several apple varieties. It has also been demonstrated that the perceptions of sweetness and tartness are not independent from each other [31,32]. The overall flavour evaluations indicated that panellists perceived lower flavours from the lowest DMC category (i.e., 12.5% ± 0.5 DMC from Cawston). In general, SuRDC fruits were evaluated as fruits with greater overall flavour compared to those from Cawston and they also had higher SSC and DMC levels as well. Moreover, the results of the flavour-related CATA questions confirmed the findings of the flavour ratings because fewer panellists (than expected) reported tropical and fresh apple flavours from the Cawston 12.5% ± 0.5 DMC category, while they reported more “bland” and “no fruity flavour” off-flavours in relation to apples from this category.

The sensory evaluations and textural measurements both indicated that apples were softer after a ten-week cold storage. Their mass shrank slightly and their TA levels dropped. Costa et al. [33] reported similar changes in apples during storage. The TA levels increased with increases in DMC levels and red over colouration in apples stored only for two weeks, as also reported by Dever et al. [28] for the relationship between red over colour and sourness flavour.

Carbohydrates comprise 90% of DMC in apple fruit [34]. About ¾ of total carbohydrates in mature apple fruit are digestible; starch content is usually negligible, whereas sugars comprise about 65–80% of DMC and determine fruit sweetness. Structural carbohydrates consist of ¼ of total carbohydrates, in which pectin and other hemicelluloses comprise 7–11% of DMC [35]. As the critical cell wall components, the nature of these large structural polysaccharides plays an important role in fruit firmness. As structural carbohydrates take up a smaller portion in DMC than sugars, this explains why firmness, which depends on pectin stability, demonstrated less relevance to DMC% in this study on the ‘Ambrosia’ apple (Figure 3C). In contrast, SSC%, which is expressed in sucrose equivalents as an indicator for sugar content, demonstrated higher correlation with DMC% (Figure 4A and Figure 5A). The dynamics of structural and non-structural carbohydrate compositions of the apple fruit are cultivar-dependent intrinsic characteristics [34,35] that are subject to the predisposition of environmental and horticultural conditions. In some varieties such as ‘Royal Gala’, DMC% demonstrated more relevance to fruit firmness [6]; this could be attributed to a greater portion of structural carbohydrates in these apples. High correlation between DMC% and SSC% found in ‘Ambrosia’ could indicate higher sugar contents in its total carbohydrates. This requires further investigation in a carbohydrate profiling and DMC study on multiple apple varieties. In addition, the strong positive correlation between DMC% and SSC% (r > 0.8) suggests the feasibility of using non-destructive near-infrared spectroscopic measurement of DMC% to estimate SSC% in a large quantity of mature ‘Ambrosia’ apples. Furthermore, apoplastic sugar contents in fruit are osmosis regulators [36]; as sugars increase, the total concentration increases and the content of free water decreases, leading to more negative osmotic potential and lower total water potential. This was supported by a moderate negative correlation between SSC% and *Ψ*_fruit_ in this study (Figure 5B). Within a certain threshold, the decrease in *Ψ*_fruit_ could be attributed to the accumulation of SSC; extreme low *Ψ*_fruit_, however, could reflect preharvest water deficit [12] and may lead to the decrease in juiciness. Further study is required to define the *Ψ*_fruit_ thresholds in relation to preharvest growing conditions and sensory perceptions.

Fruit carbohydrate contents depend on carbohydrate assimilation and partitioning at the whole-tree level, which is influenced by photosynthetic capacity and source–sink relation [37]. Favourable environmental conditions, such as sufficient water and nutrient availability, adequate canopy light interception and low night temperature, can promote photosynthesis and photosynthate accumulation; in turn, more sugars can be translocated to fruits, leading to higher SSC% and DMC%. From the perspective of fruitlet competition, sink quantity and sink strength matter the most: when there are fewer fruitlets and lower crop load, more carbohydrates are available to each fruit [7,37,38], and more carbohydrates are allocated to the fruitlets that possess more cells and stronger sink strength. Therefore, horticultural practices that influence fruitlet cell division also play important roles in determining the size, SSC% and DMC% of mature fruit. Good fruit quality attributes can be attained by moderating crop load to avoid excessive fruitlet competition, and by maintaining the king fruitlets that have the best inherent growth potential. In this study, DMC%, SSC% and malic acid equivalent contents were higher in the fruits from SuRDC, which can be attributed to higher net photosynthetic rate (mean ± SE: 18.58 ± 0.83 µmol CO_2_ m^−2^ s^−1^ for SuRDC, 16.03 ± 0.94 µmol CO_2_ m^−2^ s^−1^ for Cawston; F(1, 11) = 9.05, *p* = 0.013; data not graphed) and lower average daily minimum temperature in August (16.19 °C versus 15.68 °C) (Appendix A). In addition, thinning was conducted by hand at SuRDC to reserve the king fruitlets, whereas at the Cawston location, chemical blossom thinning may have injured king blossoms and caused king fruitlet abortion; this could lead to the distribution of more fruits with greater growth potential at SuRDC. As fruit DMC is subject to the inter-annual variabilities of climatic and horticultural conditions, a multi-year, multi-location experiment is required to evaluate fruit DMC% under the confluent effects of water and nutrient availability, air temperature, canopy light interception, tree vigor and crop load management and to identify the main determinant(s) under specific orchard circumstances.

## 5. Conclusions

The application of DMC as a sorting factor demonstrated a potential for the selection of apples with higher SSC, sweetness and flavour but fruit firmness and textural attributes were not influenced by their DMC levels. As a result, DMC can be considered alongside indicators related to other intrinsic quality attributes to improve the precision of the sorting process. Fruit mass and near-infrared spectroscopic measurement of DMC can be used together to non-destructively characterize location-specific effects of preharvest growing conditions on fruit growth potential. Future studies will benefit from a diversity of apple varieties with different SSC and firmness profiles cultivated under different environmental and horticultural conditions.

## Figures and Tables

**Figure 1 foods-12-00219-f001:**
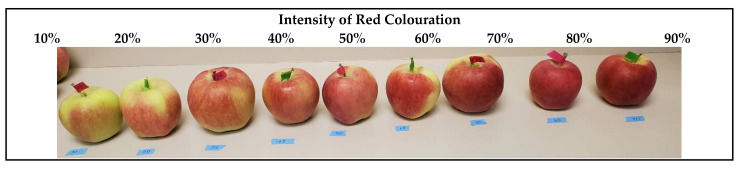
The developed scale to measure the intensity of red over colouration in percent.

**Figure 2 foods-12-00219-f002:**
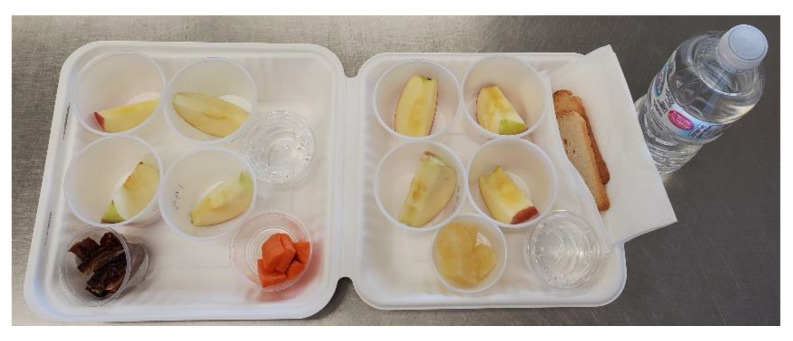
A sample tray prepared for a panellist with eight samples from four DMC categories using a randomized block design.

**Figure 3 foods-12-00219-f003:**
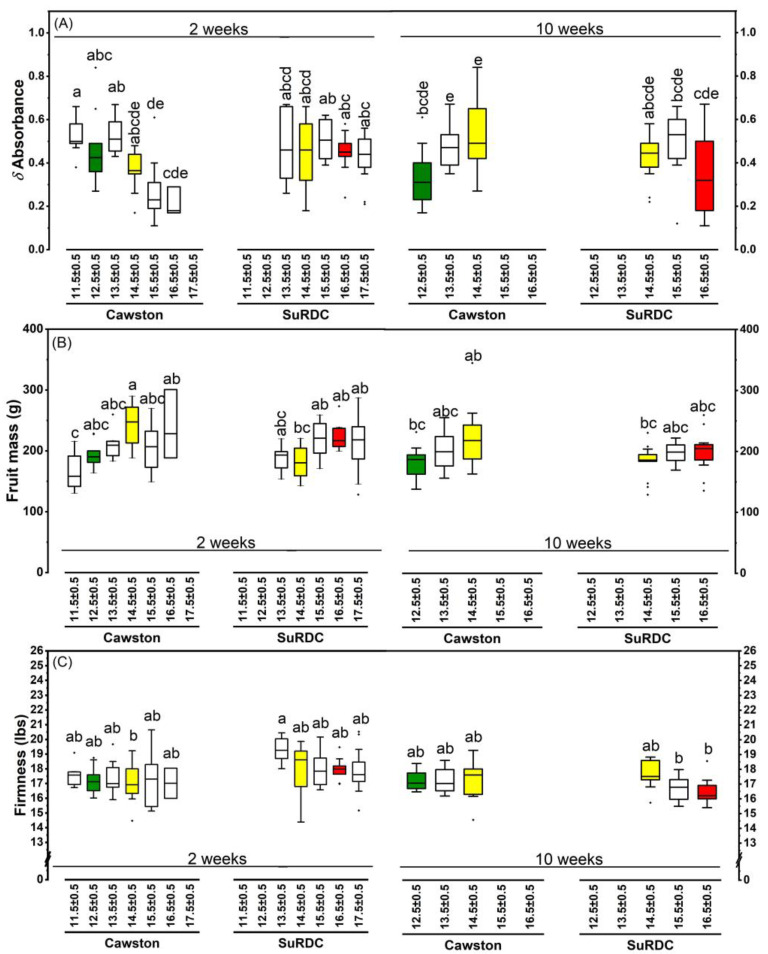
*δ*Absorbance of fruit skin (**A**), fruit mass (**B**) and fruit firmness (**C**) of ‘Ambrosia’ apples of different dry matter content (DMC%) from two locations and two storage durations (analysed on October 6 after 2-week storage and on December 2 after 10-week storage). Different lowercase letters in each sub-panel indicate significant differences among DMC levels nested in location and storage duration effects (i.e., DMC[Location × Storage duration]) at *p* < 0.05 using a three-way double-nested ANOVA. The DMC categories used in the sensory analysis are highlighted in colour, as green for 12.5% ± 0.5 for the Cawston location, red for 16.5% ± 0.5 for the SuRDC location, and yellow for 14.5% ± 0.5 on both locations.

**Figure 4 foods-12-00219-f004:**
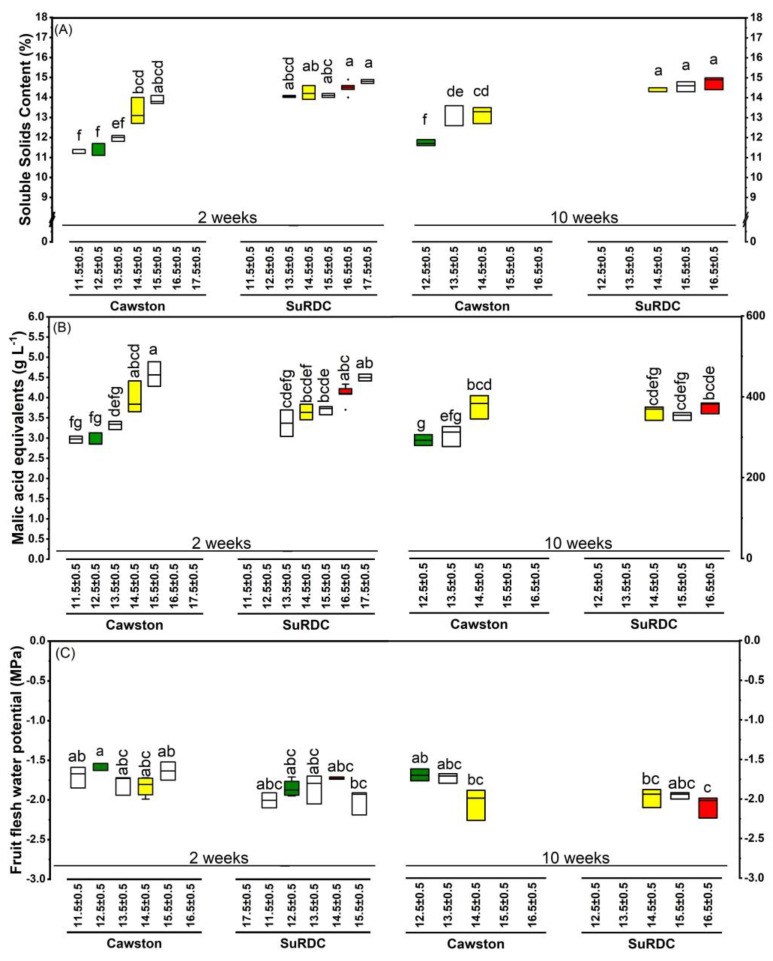
Soluble solids content (SSC) (**A**), malic acid equivalents (**B**), and fruit flesh water potential (**C**) of ‘Ambrosia’ apples of different dry matter content (DMC%) from two locations and two storage durations (analysed on October 6 after 2-week storage and on December 2 after 10-week storage). Different lowercase letters in each sub-panel indicate significant differences for DMC nested in location and storage duration effects (i.e., DMC[Location × Storage duration]) at *p* < 0.05 using a three-way double-nested ANOVA. The DMC categories used in the sensory analysis are highlighted in colour, as green for 12.5% ± 0.5 for the Cawston location, red for 16.5% ± 0.5 for the SuRDC location, and yellow for 14.5% ± 0.5 for both locations.

**Figure 5 foods-12-00219-f005:**
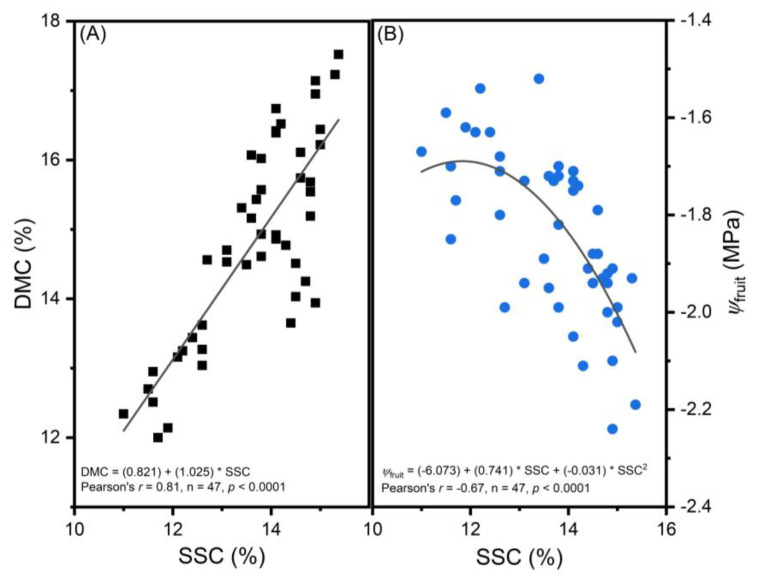
Soluble solids content (SSC%) correlated positively with dry matter content (DMC%) (**A**) and negatively with fruit flesh water potential (*Ψ*_fruit_) (**B**) of ‘Ambrosia’ apples. Linear regression in 5A and quadratic regression in 5B are shown as straight lines (n = 47, *p* < 0.001).

**Table 1 foods-12-00219-t001:** The number of fruits analysed in compositional and textural measurements in different dry matter content (DMC) categories from two locations.

Air Storage Duration	Location	DMC Category	Sample Size
**2-week**	Cawston (n = 53)	11.5% ± 0.5	9
		12.5% ± 0.5	10
		13.5% ± 0.5	9
		14.5% ± 0.5	11
		15.5% ± 0.5	11
		16.5% ± 0.5	3
	Summerland Research and Development Centre (SuRDC) (n = 56)	13.5% ± 0.5	5
		14.5% ± 0.5	10
		15.5% ± 0.5	10
		16.5% ± 0.5	10
		17.5% ± 0.5	21
**10-week**	Cawston (n = 45)	12.5% ± 0.5	15
		13.5% ± 0.5	15
		14.5% ± 0.5	15
	Summerland Research and Development Centre (SuRDC) (n = 44)	14.5% ± 0.5	14
		15.5% ± 0.5	15
		16.5% ± 0.5	15

**Table 2 foods-12-00219-t002:** Attributes, definitions and food standards for sensory evaluations.

Attribute	Definition	Scale and Food Standard
Flesh hardness	The resistance to compression when the sample was placed on the back (molar) teeth and the teeth were compressed.	Evaluated on a 100-unit unstructured line scale relative to zucchini sticks at 0 and carrot sticks at 100 units.
Juiciness	The relative juice released when compressing the apple with the back (molar) teeth.	Evaluated on a 100-unit unstructured line scale relative to snacking date at 0 and canned pineapple chunks at 100 units.
Overall texture quality	The overall impression of quality considering crispness, hardness, juiciness, skin toughness or other texture attributes.	A line scale with very poor and very good marked at 0 and 100 units
Sweetness	The taste stimulated by sugars; assessed after repeated chewing.	Evaluated on a 100-unit unstructured line scale relative to 40 g L^−1^ sucrose (Rogers fine granulated sugar; Lantic Inc., Montreal, QC, Canada) as the mid-point.
Tartness	The taste stimulated by acids; assessed after repeated chewing.	Evaluated on a 100-unit unstructured line scale relative to 0.6 g L^−1^ malic acid (Sigma-Aldrich Canada Co., Oakville, ON, Canada) as the mid-point.
Fruity flavour	Fruity flavours are tree-ripened aromatics and do not include sweet and tart flavours. There was a check-all-that-apply (CATA) question to describe the perceived fruity flavours; assessed after repeated chewing.	Using a CATA question with options including: citrus (lemon), floral/perfume (rose, jasmine, etc.), fresh apple, melon (watermelon, cantaloupe, honeydew, etc.), no fruity flavour, pear, spice (cinnamon, allspice, liquorice, etc.), stone fruit (apricot, peach, etc.), tropical (banana, pineapple, mango, etc.), and other (please specify).
Off flavours	Selecting undesirable flavours using a CATA question.	Using a CATA question with options including: alcoholic, bitter bland, cooked, green or grassy, musty or mouldy, starchy, no undesirable flavour, unripe, vinegar or acetic acid, and other (please specify).
Overall flavour quality	The overall impression of quality considering the sweetness, tartness, fruity flavour or other taste/flavour attributes.	A line scale with very poor and very good marked at 0 and 100 units.

**Table 3 foods-12-00219-t003:** The distribution of fruits in DMC categories, fruit mass, fruit red over colouration (distribution and intensity) and skin ground colour in apples from two orchards.

Location	Estimated Dry Matter Content (DMC) Category	Number of Apples in DMC Categories	Average Mass (SE) (g)	Skin Red Over Colour (Median in %)	Intensity of Red Over Colouration (Median in %)	Number of Apples with the Identified Ground Colour		
						Green	Yellow-Green	Yellow
Cawston (n = 270)	11.5% ± 0.5	9	167.40 (10.31)	70	30	3	6	0
	12.5% ± 0.5	79	189.95 (2.98)	50	40	29	40	10
	13.5% ± 0.5	106	206.43 (3.11)	60	40	39	25	42
	14.5% ± 0.5	55	223.72 (5.09)	60	50	19	12	24
	15.5% ± 0.5	17	218.77 (9.44)	90	55	1	6	10
	16.5% ± 0.5	3	244.05 (34.09)	85	70	0	1	2
Summerland Research and Development Centre (SuRDC) (n = 270)	13.5% ± 0.5	5	189.40 (11.41)	75	70	2	2	1
	14.5% ± 0.5	46	189.02 (3.59)	80	60	10	33	3
	15.5% ± 0.5	124	201.24 (2.67)	75	50	26	90	8
	16.5% ± 0.5	75	204.97 (3.43)	70	50	25	46	4
	17.5% ± 0.5	21	214.25 (8.79)	75	50	4	16	1

**Table 4 foods-12-00219-t004:** Sensory evaluations reported for apples from different DMC categories from two locations.

		Flesh Hardness		Juiciness		Overall Texture Quality		Sweetness			Tartness		Overall Flavour Quality		
Location	DMC Level	LSMean *	SE **	LSMean	SE	LSMean	SE	LSMean	SE		LSMean	SE	LSMean	SE	
Cawston	12.5% ± 0.5	56.80	3.18	63.73	2.54	65.16	3.33	49.93	3.79	b	39.39	4.50	55.52	3.62	b
	14.5% ± 0.5	62.55	3.18	64.05	2.54	70.00	3.33	56.45	3.79	a	40.91	4.50	67.25	3.62	a
SuRDC	14.5% ± 0.6	58.23	3.18	62.36	2.54	65.25	3.33	57.93	3.79	a	41.55	4.50	65.11	3.62	a
	16.5% ± 0.5	63.41	3.18	65.84	2.54	69.43	3.33	59.00	3.79	a	40.39	4.50	68.68	3.62	a
F ratio		3.15		2.10		2.38		5.18			0.31			7.11	
*p* value		0.06		0.15		0.12		0.015			0.74			0.005	

* LSMean: least square mean; ** SE: standard error. Different letters in the same column indicate significant differences (*p* < 0.05).

## Data Availability

Data from compositional and instrumental analyses are available upon request.

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
