# Peer review of "Internal Quality Attributes and Sensory Characteristics of ‘Ambrosia’ Apples with Different Dry Matter Content after a Two-Week and a Ten-Week Air Storage at 1 °C"

_foods, 2023, doi:10.3390/foods12010219_

Round 1

Reviewer 1 Report

The original manuscript titled 'Internal quality attributes and sensory characteristics of Ambrosia apples with different dry matter content after a two-week and a ten-week air storage at 1 ºC' is well written and the novelty regarding the relationship between quality parameters and dry matter content (DMC) is high. However, some suggestions could be improved:

-Line 78: Please, add one reference of the protocol for the measure with the Felix-750 Produce Quality Meter.

-Line 83: Please, add the unit of the scale in figure 1 (% or hedonic scale?).

-Lines 90 and 92: Please, add % of RH at 23 ºC.

-Line 103: N is referred to the number of replicates of one experiment, no regarding the total fruit number or sample size. Please, correct it.

-Line 107: Please, briefly explain the protocolo to measure the blush/background transition zone of the apple fruits.

-Line 114: Please, change the units to mg 100 g-1 and add one reference about the protocol to measure total acidity.

-Line 127: sensory panel judges, 12 judges in total, were...

-Line 129: Please, add information about hours of training of the judges, percentage of female and male gender and range of age. Moreover, add detail in line 136 about temperature at which samples were served.

-Table 2: g L-1.

-Table 2: Why the scale was based from 0 to 100 units? Please, add a reference of this sensory protocol.

-Line 204: visual color evaluation.

-Line 218: This result should be included in the point 3.1.2.

-Point 3.1.4.: Location could be better than site. Please, correct it throughout the manuscript.

-The visual quality of figure 3 is poor.

-In some cases, the significant level (p) is used as lowercase letter (p) and in other cases with capital letter (P). In addition, some times italics are not used. Please, correct this mistake.

-Line 310: Sensory evaluation results

-Point 3.3.7: (N = 3, 176 fruits in total).

-As you discussed in line 372, which are the pH values of the apple fruits in the present experiment? Did you measured? Please, include these results.

-In addition, the analysis of structural and non-structural carbohydrates in these samples could be interesting.

-As you well mentioned, temperature, among other climatic conditions/factors, could be influenced on DMC. In this sense, please, add an information table (as supplementary table) with the temperatures of the two locations evaluated. The values of temperature in line 451 are not understood.

-Line 465: Please, add more information about one year more of study is needed to corroborate all results and to investigate the influencing climatic factors, as you mentiones at the end of the discussion section.

Author Response

Dear Reviewer,

Thank you for reviewing our manuscript foods-2085595 and for providing valuable revision suggestions. We carefully revised the manuscript and prepared the attached document of our responses to your comments for your review.

Yours sincerely,

Masoumeh and Hao, from Summerland, Canada

Reviewer 2 Report

 This study determine the compositional and textural characteristics and sensory profile of Ambrosia apples with different dry matter content (DMC) as estimated by Felix-750 Produce Quality Meter. The article is written in sound way. There are few following suggestions for the article.

Authors may cite few more recent works where researchres has correlated DMC with quality traits in apples.
Author may include the reason to use Felix instrument instead of direct DMC measurements.
Some sentences are too long to understand easily e.g. L47-52. Such sentences may be converted into a couple of comprehensible sentences if possbile.

Author Response

(The authors gave the same response as above.)
